# GitGraph - from Computational Subgraphs to Smaller Architecture Search Spaces

**Kamil Bennani-Smires, Claudiu Musat, Andreea Hossmann, Michael Baeriswyl**
Swisscom AI Lab
Genfergasse 14, Bern, Switzerland
`{first.lastname}@swisscom.com`

## Abstract

To simplify neural architecture creation, AutoML is gaining traction - from evolutionary algorithms to reinforcement learning or simple search in a constrained space of neural modules. A big issues is its computational cost: the size of the search space can easily go above ~$10^{10}$ candidates for a 10-layer network and the cost of evaluating a single candidate is high - even if it's not fully trained.

In this work, we use the collective wisdom within the neural networks published in online code repositories to create better reusable neural modules. Concretely, we (a) extract and publish GitGraph, a corpus of neural architectures and their descriptions; (b) we create problem-specific neural architecture search spaces, implemented as a textual search mechanism over GitGraph and (c) we propose a method of identifying ***unique*** common computational subgraphs.

## 1 Better AutoML Search Spaces

Current automated neural architecture creation strategies rely on extensive expert knowledge and heavy handed supervision. They either use predefined modules Negrinho & Gordon (2017) and the novelty lies in the recombination or they create new modules but within a very tightly controled structure Zoph & Le (2016); Such et al. (2017). The reason for this heavy-handed supervision is that each step taken towards a better architecture is costly. This constraint is independent of the search method used. Whether it's employing reinforcement learning Zoph & Le (2016); Baker et al. (2016) or evolutionary algorithms Such et al. (2017), for each change the system must evaluate candidates and each evaluation means training a full network on a usually complex task. The smaller the changes, the more candidates need to be evaluated. The space of possible options is too large to allow searching or evolving a full architecture from basic building blocks like matrix additions or multiplications. Shortcuts are thus necessary.

Neural evolution can be seen as a combination of two problems - defining a neural module search space and creating a policy to create that space. The question of finding the right policy has received almost all the community's attention Negrinho & Gordon (2017); Such et al. (2017); Zoph & Le (2016), with the search space receiving almost none. Notable exceptions are Schrimpf et al. (2017), who explicitly state that different domains require different operators, that are subsequently combined to form neural architectures and Negrinho & Gordon (2017) who allow experts to state what are the modules to use for a task.

We propose constructing the search space by using the known architectures for similar tasks. Expert supervision can guide the search and lower the network creation cost. In our view, however, this supervision need not be a laborious task linked to the task at hand. Instead, it can come from repositories of computation graphs previously published for similar tasks.

As shown in figure 1, we split the task of search space definition into three parts: 1. Search for architectures that solve similar problems. This step yields a collection of graphs. 2. Common Subgraph Mining. Extract the neural modules and combinations of modules that are common between the found architectures, like *convolution + Max pool + affine*. 3. Defining the Search Space by specifying which modules are large, frequent and unique enough to be useful. These subgraphs then become a toolbox of task-oriented modules. The resulting task specific module toolbox becomes the starting point to evolve new architectures.

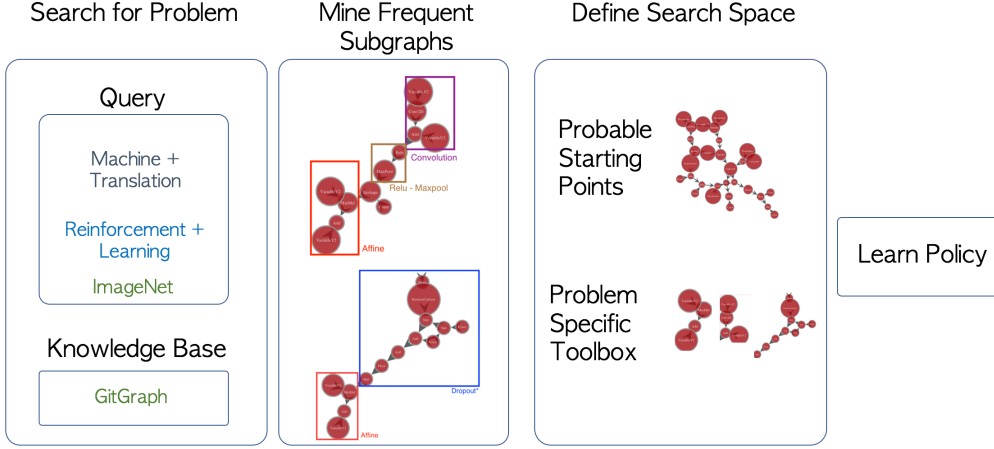

Figure 1: Automated Architecture Search Space Definition

## 2 GITGRAPH

We introduce GitGraph [1] - a dataset of TensorFlow Abadi et al. (2015) computational graphs, alongside with the description of tasks they are useful for. We checked out Github repositories of neural networks written in Tensorflow. We save the graphs from the available checkpoints $.ckpt\,files$. We then stored the graphs, alongside with the descriptions in the *readme* files and the github descriptions. We use the search functionality of the database to retrieve the graphs that are linked to specific problems or techniques, like *reinforcement learning*. A node in Tensorflow graph contains the operation performed (e.g. addition or convolution) and possibly additional information (e.g. hyperparameters values). We convert the TF checkpoint contents to Graph-tool [2] graphs and use them Yan & Han (2003; 2002) in all subsequent processing.

In its current version, GitGraph contains 6863 graphs in total, coming from 1449 repositories, for an average of 4.73 graphs per repository. Most architectures contain between $10^2$ and $10^3$ nodes, with the smallest 20% having less than $10^2$ and the largest 20% more than $10^3$ nodes. Many of the graphs are duplicates, due to multiple checkpoints for the same model and forked repositories. We remove exact graph duplicates, leading to a subset of 2033 unique graphs from the original 6863.

**Defining the scope** If, for instance, someone is interested in *machine translation for Swiss German*, we may not find any prior researchers who tackled that specific problem. A reasonable assumption is that neural architectures made for *machine translation* or, in the best case for *machine translation German* are similar to the given task. We study three tasks: image processing, text processing and reinforcement learning. GitGraph contains: (139 graphs with duplicates , 80 without) for images, (77 / 29) graphs for text and ( 283 / 88) for reinforcement.

**Graph Cleaning**. We preprocess the graphs, in order to focus on the core architecture. We remove all nodes created by the optimizer including all the subsequent gradient computation nodes; all nodes concerning saving/restoring variables as well as summarization (visualisation on the tensorboard); all nodes used to initialize a variable and we reduce multiple nodes in a variable definition to a single one. We remove assign and identity nodes, and forward the edges directly to the variable node.

## 3 FREQUENT SUBGRAPHS

The second contribution is a method of finding relevant neural subgraphs for a given task. We define a subgraph that is common at $\tau\%$ for task $T$ as one that appears in a minimum of $\tau\%$ of the graphs for task $T$. The higher the threshold, the more likely it is that the subgraph is actually relevant. For low $\tau$ values the subgraphs are uninformative; we thus set a minimum value for $\tau$ of 30%.

---

[1] https://www.mycloud.ch/s/S00E8129370EFE75830040072AD8203611E4F9971E1
[2] https://graph-tool.skewed.de

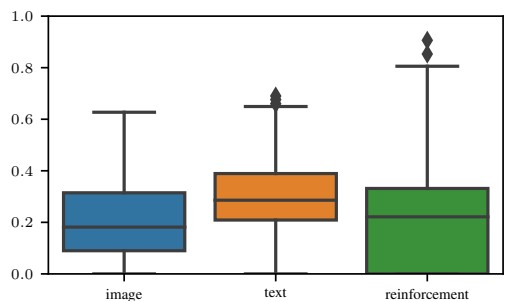

(a) Node reduction distribution on the whole data

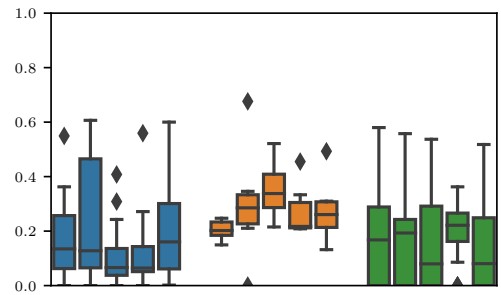

(b) Node reduction distribution on five test sets per task

Figure 2: Frequent subgraphs account for 20-30% of all the graph nodes

The bigger subgraphs are more informative than smaller ones they contain. Our solution is to not count the occurrences of subgraphs that are contained by even larger **and** common subgraphs. The reduction is significant, with roughly two thirds of the common subgraphs eliminated because of their belonging in larger units. In the special case of reinforcement learning - with $\tau = 30\%$ we obtain 23 frequent subgraphs but only 1 remains after the subgraph reduction.

The frequent subgraphs are meaningful, large chains, containing tens of nodes. If they are replaced with single nodes, they can lead to a significant reduction of the complexity of the network. The size of the common reinforcement subgraph discussed in the previous section is 30 nodes. For image and text analyses, roughly half of the frequent subgraphs have a size of between 3 and 5 nodes. The other half is between 5 to 15 for image and 5 to 30 for text. A manual analysis of these nodes shows that they correspond to commonly used recurrent units (e.g. LSTM).

These subgraphs appear a large number of times in the original graphs. Their replacement with single nodes leads to a large graph complexity reduction, defined as the aggregated size of all the frequent subgraphs, normalized by the entire size of the graphs they appear in. Figure 2 a) plots the complexity reduction for the three studied tasks. The median reduction for the three tasks ranges from 20 to 30%, with the highest value being recorded for text.

Furthermore, we test the hypothesis that the subgraphs found are general and linked to the task itself. For each task, we create five different experiments in which we randomly split the data into 80% for the training set and 20 % for the test set. We then determine whether the subgraphs mined from the training set occur in the graphs in the test set. We report the results for each task individually in Figure 2 b). The median and variance change very little from the previous experiment, upholding its conclusion. It shows the complexity reduction is achievable on unseen graphs, for the same task.

## 4 CONCLUSION

We introduced GitGraph, the first corpus of neural computation graphs. The first goal of GitGraph is to serve as a knowledge repository that allows for an automated search of neural architectures that solve a specific problem. Using the search functionality, we can obtain a set of distinct architectures for problems related to the searched one. From the found architectures, in the form of computation graphs, we created a method of generating unique relevant common subgraphs.

The main aim of finding problem-specific frequent subgraphs is to create a neural search space containing larger elements. We hope this will reduce the complexity and cost of the subsequent neural architecture creation policy by optimizing the search space itself.

We show that the GitGraph common subgraphs cover between 20 and 40% of the nodes in their source graphs. Given the obtained complexity reduction, we believe they will be a basis for large problem-specific modules in future automated neural creation strategies.

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
