# OpenReview forum: "GitGraph - from Computational Subgraphs to Smaller Architecture Search Spaces"
_ICLR.cc/2018/Workshop — Accept_

### Official Review · AnonReviewer3 · 2018-03-06
**Good workshop paper**

**Rating:** 7
**Confidence:** 3

**Review:**

The authors propose a method to reduce the search space when designing neural network architectures. A corpus of neural networks has been collected from public sources and relevant task-specific building blocks are identified via frequent subgraph mining.

Designing effective network architectures requires tedious manual work. Therefore, the authors address an important topic. Although many questions remain open, I really like the approach to address this by mining published network architectures.

Strong points:
  * Important subject
  * Nice and original idea
  * Some preliminary results
  * Well writen

In summary, I think the article presents a nice and relevant contribution and should be accepted for the workshop.

---

### Official Review · AnonReviewer1 · 2018-03-09
**GitGraph - From Computational Subgraphs to Smaller Architecture Search Spaces**

**Rating:** 4
**Confidence:** 4

**Review:**

In this paper, the authors propose to create a database of frequently used computational subgraphs by mining publicly available TensorFlow implementations of deep learning models in GitHub. Their vision is that such a database could serve as a starting set of modules for neural architecture search algorithms.

Neural architecture search and, more generally, AutoML, is definitely a topic of great interest for the community. As such, efforts to explore and better understand existing successful architectures are definitely worthwhile.

While the idea proposed in the article is interesting, I have some major concerns regarding the way in which the database was created.

Firstly, it appears that the authors did not apply any quality control mechanisms when selecting which GitHub repositories should be incorporated into the database. This means that, in effect, an implementation of a certain model written by a hobbyist or an undergraduate student trying to solve a homework problem is given the same relevance than repositories of papers subjected to peer-review or which are the result of large, coordinated development efforts by institutions with a proven track record of reliability. Most importantly, nothing seems to prevent their system from incorporating incorrect or incomplete implementations into their database.

This problem is compounded by the fact that the authors define the relevance of a subgraph entirely based on its frequency in the corpus: this could skew the distribution of “relevant” computational subgraphs found to those which are easier to use or simply more popular rather than those which account for performance improvements over a certain meaningful baseline.

Instead, I strongly believe that the authors should: (1) think of quality control criteria to decide which repositories should be included and which ones should not and (2) give weights to repositories depending on how well they perform in a given task, and use such metric to substitute the mere frequency count they are currently using.

Finally, while the main motivation of the paper seems to stem from AutoML, the authors provide no compelling evidence that their dataset can be used to improve existing neural architecture search algorithms.


Minor points:

The text inside nodes of the computational graphs in Figure 1 is barely legible.

The choice of setting the support threshold to 30% seems arbitrary and should be better justified.

Keeping only frequent subgraphs which are not contained by other frequent subgraphs is referred to as “maximal frequent subgraph mining” in the data mining literature.

Typos

Abstract: “A big issue(s) is its computational cost”

---

### Official Review · AnonReviewer2 · 2018-03-10
**Good idea to create a dataset but no strong results yet.**

**Rating:** 7
**Confidence:** 4

**Review:**

In this paper, the authors show how they create a dataset of neural network graphs from online repositories. This is an interesting dataset and the authors use it to mine some frequent subgraphs. While a possible use of the dataset, one could imagine more baselines and wish for stronger results. Nevertheless, it is a nice dataset.

---

### Decision · Program_Chairs · 2018-03-20
**ICLR 2018 Workshop Acceptance Decision**

**Decision:**

Accept

**Comment:**

Congratulations, your paper was accepted to the ICLR workshop.